# Quality of Work Life and Contribution to Productivity: Assessing the Moderator Effects of Burnout Syndrome

**DOI:** 10.3390/ijerph18052425

**Published:** 2021-03-02

**Authors:** João Leitão, Dina Pereira, Ângela Gonçalves

**Affiliations:** 1Faculty of Social and Human Sciences, Research Center in Business Sciences (NECE), University of Beira Interior, 6200-001 Covilhã, Portugal; 2Research Center in Business Sciences (NECE), University of Beira Interior, 6200-209 Covilhã, Portugal; dina@ubi.pt (D.P.); angela.goncalves@ubi.pt (Â.G.); 3Centre for Management Studies of Instituto Superior Técnico (CEG-IST), University of Lisbon, 1049-001 Lisboa, Portugal; 4Instituto de Ciências Sociais (ICS), University of Lisbon, 1649-004 Lisboa, Portugal

**Keywords:** burnout, emotional exhaustion, low effectiveness, cynicism, quality of work life, productivity

## Abstract

This study is focused on assessing the effects of burnout as a moderator of the relationship between employees’ quality of work life (QWL) and their perceptions of their contribution to the organization’s productivity by integrating the QWL factors into the trichotomy of (de)motivators of productivity in the workplace. The empirical findings resulting from an OLS multiple regression, with interaction terms, applied to a survey administered at 514 employees in 6 European countries, point out two important insights: (i) QWL hygiene factors (e.g., safe work environment and occupational healthcare) positively and significantly influence the contribution to productivity; and (ii) burnout de-motivator factors (that is, low effectiveness, cynicism, and emotional exhaustion) significantly moderate the relationship between QWL and the contribution to productivity. Combining burnout with other QWL components, such as occupational health, safe work, and appropriate salary, new insights are provided concerning the restricting (i.e., low effectiveness and cynicism) and catalyzing (emotional exhaustion) burnout components of contribution to productivity. These findings are particularly relevant given the increased weight of burnout, mental disorders and absenteeism in the labor market, affecting individuals’ quality of life and organizations’ performance and costs.

## 1. Introduction

The lack of quality of work life (QWL) is associated with higher levels of work-related occupational stress, anxiety and burnout, which lead to lower job performance and induces significant costs for organizations [1].

Recently, the World Health Organization (WHO) categorized burnout as a job-related phenomenon [2], characterized by chronic stress [3]. Burnout embraces three main dimensions, emotional exhaustion or energy reduction, the negative emotional state of cynicism and low professional effectiveness [2,4,5,6,7,8,9,10]. The costs of burnout are growing fast [8,11], affecting currently 13–25% of the working population [12].

Workers with a high level of emotional intelligence usually have reduced burnout levels [13] as these individuals are more able to deal with stress, which in turn could lead to a higher level of productivity [14]. Emotional regulation techniques can aid individuals to feel accomplished at work [15]. As a sense of low accomplishment is one of the dimensions of burnout, emotional regulation strategies could help to prevent this syndrome.

Such a sense of not being able to accomplish duties is associated with the lack of workers’ health and wellbeing, which in turn can be caused by poor working conditions [16]. A related OECD working paper states that health is an important factor in the relationship between work factors and productivity, and strong evidence was found of a negative relationship between job stress and productivity [17].

According to available data, as a whole, from 2018, productivity has been increasing since 1995 in virtually every EU country [18], although its pace growth has been slowing down. Increasing the productivity of individuals and organizations is among the essential objectives of the Europe 2020 strategy for growth, driven by international competitiveness concerns and the promotion of productivity, growth and sustainability [19], and more recently, with the reinforcement of quality of life through truly green and sustainable growth, as contained in the European Green Deal [20].

Following such a need to improve workers’ productivity, the QWL requires further reinforcement in order to spur employees’ motivation which is increasingly important in the context of digital transition observed in highly skilled and technologically advanced economies [21]. Adding to the previous statements, the QWL’s improvement is also in line with the worldwide commitment for accomplishing the 17 sustainable development goals (SDGs), as defined by the United Nations.

As previously outlined by [15], besides emotional regulation techniques, there is a need for further research to extend knowledge about burnout components or (de)motivators and their role in accelerating individuals’ personal commitment to organizations’ performance, incorporating the still limited knowledge on the components of QWL as a cornerstone of organizational performance.

In this line of reasoning, the current study provides an innovative assessment of the effects of burnout as a moderator of the relationship between employees’ QWL and their perceptions of their contribution to the organization’s productivity, highlighting the integration of the QWL factors into the trichotomy of (de)motivators of productivity in the workplace. By doing so, it makes a two-fold contribution, namely: (i) testing the moderating effects of burnout on QWL, disaggregating the interaction effects by motivators and hygiene per component of QWL; and (ii) revealing the burnout components, as de-motivators, which restrict or catalyze the relationships between distinct components of QWL and employees’ contribution to productivity.

This paper is structured as follows: first, a literature review and the hypotheses are presented, followed by the research methodology. Then the results are discussed and conclusions drawn, ending with the limitations and implications of the study.

## 2. Theoretical Background and Research Hypotheses

### 2.1. Quality of Work Life and Productivity: From the Roots into an Integrating Model

Recovering the roots of the guiding literature on job satisfaction originated in QWL, Herzberg proposed a model where the factors involved in attaining job satisfaction are different from the factors that prompt job dissatisfaction [22]. Herzberg asserts that the opposite of job satisfaction is not job dissatisfaction, but instead no job satisfaction [23].

This theory, known as Herzberg’s two-factor theory or motivation-hygiene theory, proposes that motivation factors are intrinsic to the job while hygiene factors are extrinsic to the job. The motivators or growth factors are achievements, recognition, the work itself, responsibility and advancement, while the hygiene or dissatisfaction-avoidance factors are: company policy and administration, supervision, interpersonal relationships, working conditions, salary, status and security [22,23]. The results of Herzberg’s work indicate that motivators are the cause of job satisfaction and hygiene factors the cause of unhappiness on the job [23]. This theory was the subject of many scientific studies, some supporting this theory [24,25,26,27,28], while others counter it [29,30,31,32].

Moreover, more recently, another theory arose, the trichotomy of motivator factors in the workplace [33], based upon Herzberg’s theory [22] and also the theory of tourist motivation factors [34]. The trichotomy of motivator factors adds another factor to Herzberg’s two-factor theory and identifies three factors involved in job satisfaction: motivators, hygiene factors and de-motivators. This theory identifies as motivators for job satisfaction the following factors: bonuses, promotion opportunities, personal development opportunities, flextime, cafeteria benefits, recognizing merit and training paid by the employer [33]. As hygiene factors, Koziol et al. identify compensation, working hours, workload, interpersonal relations, friendly atmosphere at work, industrial safety, work content, company policy, responsibility and social scheme activities. For the final and added de-motivator factors, the following are described: mobbing by superiors/coworkers, stress at work, work exceeding employee’s psychophysical potential and qualifications, short-term contracts, employer’s continuous and close supervision and lack of possibility of changing status quo/making improvements [33]. The author assumes the motivation factors represent the stimulants, the hygiene factors represent the nominants, while the de-motivator factors represent the denominants [33]. In addition, the author considers that in order to make improvements to the motivation system, the de-motivators (denominants) should be eliminated, the hygiene factors (nominants) should be optimized, and lastly, the motivators (stimulants) should be maximized [33].

With the motivation of designing an integrating model of the QWL factors into the trichotomy of (de)motivators of productivity in the workplace, it should be stressed that albeit there is a vast amount of literature on the subject of QWL, many researchers agree that QWL is different from job satisfaction and that it deals with employees’ wellbeing [35,36,37,38,39].

Hackman and Oldham proposed a model in which the needs of psychological growth (skill variety, task identity, task significance, autonomy and feedback) were connected to QWL [40].

Walton (1980) considered eight conceptual categories in QWL [41]: adequate and fair compensation; safe and healthy working conditions; immediate opportunity to use and develop human capacities; opportunity for continued growth and security; social integration in the work organization; constitutionalizing the work organization; work and the total life span; and the social relevance of work-life.

According to Sirgy et al. [40], QWL can be expressed by the satisfaction of a set of employee needs in relation to resources, activities and outcomes associated with their participation at the workplace [39].

Considering employee’s personal experiences, Martel and Dupuis (2006) define QWL as corresponding to the conditions experienced in the dynamic pursuit of one’s own hierarchically organized goals within work domains [42]. Thus, reducing the gap that separates the individual from these goals will have a positive impact on the individual’s general quality of life, organizational performance, and consequently on the overall functioning of society.

QWL can have distinct meanings based on individual perceptions, varying according to age, position in industry and career stage [43]. In addition, it should be noted that good QWL enhances wellbeing and satisfaction in the workplace [44].

According to Mejbel, Almsafir, Siron, and Alnaser (2013), the most common drivers of QWL are reward, benefits, compensation, career development, communication, safety, security, management involvement, the cohesion of work and life, job satisfaction and employee motivation [45].

As these examples from the literature of reference reflect, QWL is a multi-dimensional construct and can be described as a favorable working environment that supports and promotes satisfaction by providing employees with job security, growth opportunities, promotion, compensation and recognition [46]. QWL is associated with health, wellbeing, job security, job satisfaction, work-life balance, motivation, productivity and competence development [44,46], and it encompasses four main components: safe work environment, occupational health care, appropriate working time, and appropriate salary [47]. A poor working environment (e.g., poor safety and health, work pressure and stress) can also affect QWL, although in a negative way [48]. In fact, the work environment has been consistently reported as the most influential factor of QWL [49]. This is also in line with the previous findings of Leitão, Pereira and Gonçalves (2019), who underlined the importance of factors related to workers having their supervisor’s support, being integrated into a good work environment and feeling respected, acting as positive influencers of QWL [50].

It is suggested that organizations should provide employees with a more secure work environment so that they can perform at their best level [51]. For organizations, a good QWL has been regarded as an essential tool to attract and retain employees [52,53,54]. Furthermore, QWL is important for organizations to achieve growth and profitability, obtaining more efficient and effective outcomes from employees [55].

The core objective of QWL in an organization is to improve the employee’s wellbeing and productivity [37,56]. An organization cannot get efficient and effective outcomes from its employees without QWL since the latter is important for employees and necessary for the organization to attain growth [55]. Good management of QWL makes the organization’s employees healthier, more committed, working and producing more and better [57]. Other studies have revealed positive correlations between QWL and productivity [53,58,59,60].

Considering the above-mentioned literature, the following research hypothesis is formulated:

**Hypothesis** **1**(**H1**)**:**
*QWL’s motivating factors have a positive relationship with the contribution to productivity*.

**Hypothesis** **1a**(**H1a**)**:**
*Appropriate working time has a positive and significant effect on the contribution to productivity*.

**Hypothesis** **1b**(**H1b**)**:**
*An appropriate salary has a positive and significant effect on the contribution to productivity*.

**Hypothesis** **2**(**H2**)**:**
*QWL’s hygiene factors have a positive relationship with the contribution to productivity*.

**Hypothesis** **2a**(**H2a**)**:**
*A safe work environment has a positive and significant effect on the contribution to productivity*.

**Hypothesis** **2b**(**H2b**)**:**
*Occupational healthcare has a positive and significant effect on the contribution to productivity*.

### 2.2. Burnout and Organizational Stress

The term burnout first appeared in the pioneering study by Freudenberger in 1974 [61]. While working as a psychoanalyst, he described his own experience as a combination of feelings, exhaustion and fatigue, a lingering cold, headache and gastrointestinal disturbances, sleeplessness and shortness of breath. Despite being first mentioned almost half a century ago, burnout is still a problem and is increasingly discussed. As the discussion about burnout increases, so does the use of the word as a catchphrase, an expression that includes a variety of conditions and symptoms [62] and steers away from the original meaning and purpose.

Despite the pioneering concept being introduced by Freudenberger (1974), the earliest accepted definition of burnout was only widely spread by Maslach, Jackson and Leiter (1996), reconceptualizing burnout as the syndrome of reduced personal accomplishment, increased emotional exhaustion and increased depersonalization experienced by individuals working closely with people [63]. Concerning the origin of this type of syndrome, the burnout results from experiencing chronic stress at the workplace have not been dealt with correctly [3,64].

In 2019, the World Health Organization (WHO) updated the definition of burnout and re-characterized it as a job-related phenomenon instead of a health or mental disorder. Burnout is now defined by WHO (2019) as: “a syndrome conceptualized as resulting from chronic workplace stress that has not been successfully managed. It is characterized by three dimensions: feelings of energy depletion or exhaustion; increased mental distance from one’s job, or feelings of negativism or cynicism related to one’s job; and reduced professional efficacy. Burn-out refers specifically to phenomena in the occupational context and should not be applied to describe experiences in other areas of life”.

Job burnout can be differentiated in terms of (i) emotional exhaustion; (ii) depersonalization; and (iii) lack of personal accomplishment [4,5,6,7,8,9,10,65], and is found mostly in people who have social professions, such as teachers, doctors and social workers [64]. For example, in the USA, burnout is more common among physicians than among other workers [66].

Despite affecting professional life, burnout has also been said to affect personal life [4,5,65,66] and employees’ general health by increasing the possibility of developing sleep illnesses, obesity, diabetes, increased cardiovascular risk, faster aging, fatigue, low self-esteem, anxiety and depression [11,67]. Burnout also has been associated with suicidal tendencies and substance abuse [8,11,61]. Nonetheless, the main symptoms associated with and observed in burnout patients are chronic fatigue, continuous exhaustion, concentration disturbances, memory lapses, disorganization, lack of drive, personality changes, anxiety, depression and a low sense of personal accomplishment [11,66,67,68]. Somatic symptoms also occur and can appear in the form of headaches, gastrointestinal disorders and cardiovascular disturbances (e.g., tachycardia, arrhythmia and hypertonia) [68,69,70]. In its turn, it has been reported that smoking could have a protective effect against burnout, justifying that the reason could be that smokers take more breaks [71].

Regarding the professional aspect, job burnout has been associated with absenteeism, decreased productivity, organizational commitment, motivation and satisfaction [3,5,10]; reduced physical and mental health and affecting the quality of work [72]. In fact, the level of satisfaction in the workplace is found to have a decisive influence on workers’ health [73]. Lower levels of burnout are found in people with a greater interest in their jobs [72]. On the other hand, high levels of burnout were reported as possibly indicating a negative attitude towards work and oneself, lack of interest and lack of satisfaction with one’s work [74]. Indeed, in countries with higher burnout levels, people do not feel happy, are not satisfied with their jobs and do not feel engaged at work [75]. Job burnout has been stated as having a possible adverse impact on nurses’ performance, work satisfaction and QWL [76]. It also has been reported that job burnout is also present among academics, in particular those who belong to public universities [77]. On the one hand, a high QWL has been associated with greater productivity at the workplace [50]. On the other hand, work-related occupational stress, anxiety and burnout are related to lower job performance [78] and lead to significant costs for organizations [1]. In organizational contexts, the costs of burnout are growing fast [8,11], affecting 13–25% of the working population [12]. In addition, more burnout cases occur in countries where economic performance is lower, and higher levels of burnout are observed in countries with lower GDP and longer working hours [75].

Nowadays, the labor force is subject to increased strain due to the uncertainty, competitive climate and job insecurity that decreases wellbeing and can contribute to converting employee commitment into burnout [5,79,80].

Schaufeli (2018) suggests that burnout should not only be seen as an individual psychological state but also as a collective phenomenon with economic and sociocultural ramifications at the national level. To combat this, organizations have already started to recognize burnout as an organizational issue and are trying to promote teamwork and improve the sense of community in order to encourage commitment [8,11].

A highly stressful environment cultivates higher burnout, as staff stress is a positive predictor of burnout, as previously shown in the literature [2,3,64,81]. Occupational stress can have a negative impact on the worker’s productivity [82].

Both emotional exhaustion and burnout are considered extreme forms of stress, and as the former suggests energy depletion [83], a decrease in productivity is expected [84]. Singh (2000) identified a negative impact of burnout on productivity, mainly in terms of work quality rather than quantity [85]. Wright and Bonett (1997) also reported a negative association between emotional exhaustion and productivity, the former being the primary dimension of burnout predicting job performance [86].

Seligman and Schulman (1986) analyzed the relationship between optimism/cynicism at work and productivity, reporting higher productivity among optimists than pessimists [87]. The same authors found that pessimists seem to leave their jobs twice as frequently as optimists.

Regarding the low personal accomplishment associated with burnout, Nayeri, Negarandeh, Vaismoradi, Ahmadi and Faghihzadeh [88] identified a positive and significant relationship between personal accomplishment and productivity. The authors also found that employees with low levels of personal accomplishment only achieved low to intermediate levels of productivity. Conversely, employees with high levels of personal accomplishment achieved high to very high levels of productivity. Considering the previous literature, the following hypotheses are derived:

**Hypothesis** **3**(**H3**)**:**
*A burnout’s de-motivating factors moderate the relationship between the QWL’s motivators and hygiene factors and the contribution to productivity*.

**Hypothesis** **3a**(**H3a**)**:**
*Emotional exhaustion restricts the relationship between the QWL’s motivators and hygiene factors and the contribution to productivity*.

**Hypothesis** **3b**(**H3b**)**:**
*Feelings of cynicism restricts the relationship between the QWL’s motivators and hygiene factors and the contribution to productivity*.

**Hypothesis** **3c**(**H3c**)**:**
*A sense of being less effective restricts the relationship between the QWL’s motivators and hygiene factors and the contribution to productivity*.

### 2.3. Design of the Operational Model of Analysis

The literature review reflects that QWL is associated with health, wellbeing, job security, job satisfaction, work–life balance, motivation, productivity and competence development [44,46], including four main aspects: safe work environment; occupational healthcare (the QWL’s hygiene factors); appropriate working time; and appropriate salary [47] (the QWL’s motivating factors); and revealing that burnout (the burnout’s de-motivating factors) can be associated with: emotional exhaustion; depersonalization; and lack of personal accomplishment [4,5,6,7,8,9,10,65]. Considering the small number of studies on the moderating effect of burnout on QWL, the current study pays special attention to the moderator effects of burnout de-motivator factors in terms of the relationship between QWL and contribution to productivity. Figure 1 below, bearing in mind previous studies and the research hypotheses originating from the literature review, proposes an operational model of analysis.

## 3. Methodology

### 3.1. Sample

The study comprehends the analysis of the responses to a survey funded on different questionnaires previously used to carry out related surveys on health and wellbeing in the workplace, including the pioneering measure on the quality of work life developed by Sirgy et al. [39] and the set of analytical tools surveyed and empirically operationalized by Leitão et al. [50].

The survey was conducted from April to July 2018. A total of twelve project partners originating from Italy, Bulgaria, Cyprus, Portugal, Greece and Spain participated in data collection by interviewing employees. The intention was not to interview company owners or general managers to avoid bias in the responses. A convenience sample based on a random selection procedure was used. In each organization, a contact person was identified to ensure completion of the questionnaire, which was afterward validated by the research team.

The questionnaires were applied through personal interviews to ensure a maximum response rate. The partners followed a set of instructions for selecting interviewees: 15 companies among micro, small and medium-sized firms (10% of interviewees for each category—EU definition of SME), plus five among large firms and public entities, involving two employees per organization and totaling 514 questionnaires.

This survey made it possible to identify several factors that are potential influencers of the desire of employees to contribute (or not) to organizational productivity (Leitão et al., [50]). Furthermore, it raised unexplored factors related to the stress and the physiological and psychosomatic condition of employees, as well as their linkages with the role played by environmental and health conditions at the workplace, in promoting wellbeing at the workplace as an organizational lever for increasing satisfaction, productivity and performance.

### 3.2. Measures and Preliminary Data Analysis

Table 1 below presents the sample characterization, showing that the respondents were distributed by gender as follows: 48% women; and 52% men. Regarding employee age: 9% are aged between 20 and 25; 34% between 26 and 35; 37% between 36 and 45; 14% between 46 and 55; and only 7% are older than 55. Concerning respondents’ marital status, 35% are single, 59% are married, and almost 7% are in another situation. Regarding respondents’ role in the organization, 18% say they occupy a managerial role, 67% a qualified role and 16% a non-qualified position. Regarding education, 51% have a university degree, 22% have a post-graduate degree, 19% completed secondary education, 7% completed 9 years at school, and only 1% completed 4 years. Concerning the sector of activity of the respondents’ organizations, almost 2% belong to the primary sector, 14% the secondary, 77% the tertiary and 7% are from public organizations. Most respondents work in micro, small or medium-sized firms, 26% in microsized with 1 to 9 employees, 39% in small-sized with 10 to 49, 15% in medium-sized with 50 to 249, 14% in large companies with 250 to 1000 and only 6% in companies with over 1000 employees. Concerning the age of organizations, 16% are between 1 and 6 years old, 34% between 7 and 15, 25% between 16 and 29, almost 17% between 30 and 49 years and almost 8% have been in existence for more than 50 years. Concerning respondents’ contract type, 68% say they have a permanent contract, 11% a contract for a stipulated period, almost 9% were temporary, 5% were freelancers, and 9% reported another type of contract. Finally, respondents were asked about their qualification inside the firm, with almost 7% identifying themselves as senior managers, 10% intermediary managers, almost 17% staff in charge, 21% highly qualified employees, approximately 25% qualified, 6% semi-qualified and 8% non-qualified. In addition, 3% answered they were apprentices, and 1% said they did not know.

In this study, the dependent variable used is a contribution to productivity, as respondents were asked to what degree they feel they contribute to the organization’s productivity. The independent variables used all regarding the different aspects of QWL, such as safe work environment, occupational healthcare, appropriate working time and appropriate salary. The variables used as moderators concern burnout: emotional exhaustion, cynicism and low effectiveness.

Table 2 below presents the descriptive statistics. It can be observed that 80% of respondents feel they contribute to their organization’s productivity. The majority of respondents, 65%, feel that they have a safe work environment. Half the interviewees feel that their working time and salary are appropriate. It can also be observed that 37% of respondents reported emotional exhaustion, 20% reported cynicism, and 23% stated that they feel their effectiveness is low. In addition, the skewness and kurtosis statistics indicate a normal distribution of the variables studied. In addition, the variance inflation factors (VIF) do not indicate any potential problems of multicollinearity since they show a low average value of 2.39, which allows the subsequent ordinary least squares (OLS) regression analysis.

The pairwise correlations in Table 3 below reveal interesting associations between the contribution to productivity and three variables, that is, safe work environment, university education, and permanent contract, in a positive and significant way. On the other hand, a negative and significant association can be observed between the contribution to productivity and the feeling of low effectiveness. The safe work environment variable is negatively and significantly correlated with the variables of emotional exhaustion, cynicism and low effectiveness. Conversely, the safe work environment variable is positively and significantly associated with the variables of manager role, university education and permanent contract. The variable of occupational healthcare is significant and positively correlated with the variables of appropriate working time and appropriate salary. The appropriate working time variable is positively and significantly correlated with the appropriate salary variable and negatively and significantly correlated with the manager role. The appropriate salary variable is positively and significantly correlated with the female variable. Emotional exhaustion is positively and significantly correlated with cynicism and low effectiveness. Cynicism is positively and significantly correlated with low effectiveness and being female. Low effectiveness is positively and significantly correlated with being married.

### 3.3. Model Specification

The main reason for using OLS models is that the dataset analyzed follows a normal distribution, considering a dependent variable represented in binary terms, which allows determining the probability of the influence of a hypothetical set of independent variables arising from the literature review and the operational model of analysis proposed above. Therefore, the dependent variable takes the value of 1 when employees state they feel they contribute to productivity and 0 otherwise. Model 1 corresponds to the basic model specification. Models 2 to 4 represent the expanded model specifications, including interaction terms, to test the hypothetical moderator effects of various burnout components on QWL. Model 5 conjugates the 4 previous models’ specifications. The OLS regression model takes the usual form of:Yi = β0 + β1 Xi1 + β2 Xi2 + … + βk Xik + εi(1)
where: Yi takes the i’s value on the outcome variable; β0 is the regression constant; Xij takes i’s to score on the jth of p predictor variables in the model; j is the predictor j’s partial regression weight, and εi is the error for case i.

Using matrix notation, Equation (1) can be represented as:Y = Xβ0 + ε(2)

Y being an × 1 vector of outcome observations; X an × (p + 1) matrix of predictor variable values (with a column of ones for the regression constant); and ε is an × 1 vector of errors; where n is the sample size and p is the number of predictor variables. The p partial regression coefficients in β inform about each predictor variable’s unique or partial relationship with the outcome variable.

## 4. Results

The results of the estimation process are presented in Table 4 below. Regarding the results of the first OLS regression (model 1), where the contribution to productivity was used as the dependent variable, we find that a safe work environment and occupational healthcare (i.e., QWL hygiene factors) positively and significantly influence the contribution to productivity, while low effectiveness (e.g., a burnout component) negatively influences the contribution to productivity. Moreover, the control variables used reveal that university education has a positive and significant influence on the contribution to productivity.

Concerning the analysis of model 2, where we studied the influence of burnout de-motivator factors, namely the emotional exhaustion on components of QWL related to the contribution to productivity, this confirms the previously identified positive and significant influence of both. Interestingly, when looking at the interaction between emotional exhaustion and appropriate salary (a QWL’s motivator factor), the contribution to productivity increases. The same significant and positive sign is found concerning university education. QWL hygiene factors, safe work environment and occupational healthcare on the contribution to productivity. Once more, low effectiveness per se has a negative and significant influence on the contribution to productivity.

Looking at model 3, when assessing the influence of cynicism on the components of QWL hygiene factors, again, both a safe work environment and occupational healthcare have a significantly positive influence on the contribution to productivity. Cynicism shows a positive association with the dependent variable, whereas low effectiveness reveals a negative and significant association. Remarkably, cynicism combined with a safe work environment significantly restricts the contribution to productivity. Again, the same reported positive and significant association for university education is found.

Now observing model 4, which tests the influence of low effectiveness on QWL components in relation to the contribution to productivity, a safe work environment influences the contribution to productivity positively; again, the QWL hygiene factors showing an important effect. Moreover, from the control variables used, it can be seen that university education has a positive influence on the contribution to productivity.

Considering the last OLS model, model 5, where we considered the influence of both burnout de-motivator factors and QWL components on the contribution to productivity, the QWL hygiene factors, safe work environment and occupational healthcare influence the contribution to productivity positively. It is observed that the combinations of (i) emotional exhaustion and occupational health; (ii) cynicism and safe work environment; and (iii) low effectiveness and appropriate salary; restrict the contribution to productivity. On the other hand, having emotional exhaustion in combination with an appropriate salary is able to catalyze the contribution to productivity.

## 5. Discussion

According to the results obtained, all OLS models (1–5) give support to H2, stating that the QWL’s hygiene factors have a positive and significant effect on the contribution to productivity, and this is in line with previous literature [51], which states that a safer work environment will make employees perform at their best level, also in line with the previous findings of Kiriago and Bwisa (2013) and Leitão et al. (2019) [48,50], who found a negative correlation between a poor work environment and QWL. This agrees with previous work that considers health an important component of QWL [41,44,46]. Furthermore, as Herzberg’s work shows, not having balanced hygiene factors causes no job satisfaction [48]. In addition, the results obtained are aligned with prior theories suggesting that the hygiene factors (i.e., nominants) should be optimized in order to achieve satisfaction on the job [59], being the latter associated with employees’ sense of contribution to productivity. This is also aligned with previous works defending that QWL can be expressed through the satisfaction of a set of employee needs, which is associated with their participation, contribution at the workplace [29]. No significant direct effects related to hypothesis 1 were found.

According to the results obtained with model 1, the H2a and H2b cannot be rejected. Notably, regarding th Herzberg’s hygiene factors, having a safe work environment and benefiting from occupational healthcare schemes is able to catalyze the contribution to productivity. The same support is found in models 2, 3 and 5. In model 4, it is found to support only for H2a.

The current study raised and tested a set of hypotheses around a burnout’s de-motivator factors (that is, emotional exhaustion, cynicism, and sense of low effectiveness) and their moderator role on the relationship between the QWL’s motivators and hygiene factors, and the contribution to productivity. In fact, it is found support for H3a (in models 2 and 5), H3b and H3c. (in models 3 and 5). Indeed, benefiting from an appropriate salary, schemes of occupational health (QWL’s motivators) at work coupled with emotional exhaustion restricts the relationship between QWL and contribution to productivity (H3a), which is aligned with previous work [11,66,67,68,78,87]. Highly stressful working environments are associated with a higher burnout, being staff’ stress a positive predictor of burnout, as prior studies already conveyed [2,3,64,81]. Both emotional exhaustion and burnout impact negatively on the worker’s productivity [82], being extreme forms of stress, causing energy depletion [83], and thus slowing productivity [84,85,86]. The results now obtained ratify previous findings, connecting QWL’s motivators and hygiene factors with emotional exhaustion (the primary dimension of burnout), pointing out a negative impact of burnout on productivity.

Moreover, the empirical evidence now obtained signals that cynicism, a form of pessimism, restricts the relationship between the QWL’s motivators and hygiene factors, and the contribution to productivity, thus supporting H3b. Such results are in line with prior literature, which addressed the relationship between optimism/cynicism at work and productivity, concluding that optimist workers are more productive at work than pessimists [87]. In addition, cynic and pessimist workers are more absent than optimists are. The current study underpins a negative and significant effect of the burnout’s de-motivator factor, cynicism, as well as its moderator role on the relationship between the QWL’s hygiene factors (such as having a safe workplace) and the contribution to productivity (as found in models 3 and 5). Optimistic workers positively affect the effect of hygiene factors, such as safe work environments, on the sense of job productivity.

Taking as reference the results obtained in model 5, there is support for H3c, as there is evidence of a negative and significant effect of the burnout’s de-motivator factor, low effectiveness of the workers, and its moderator role on the relationship between the QWL’s motivators (such as having an appropriate salary), and the contribution to productivity. This is aligned with previous literature [87,88] that advocated a positive influence of personal accomplishment on productivity and suggesting that workers that feel low personal effectiveness, which this associated with burnout, are less productive on the job. The empirical findings now obtained also unveils that low effectiveness, when combined with appropriate salary, restricts the contribution to productivity. These results highlight the importance of implementing a set of initiatives for fighting job-related burnout, as in terms of costs, it is estimated that it affects between 13 and 25% of the working population [8,11], influencing employees’ performance and thus firms’ productivity. From the current findings, it is also possible to provide lines of action, namely designing appropriate payment schemes, including benefits programs targeted to personal expenses cut for the employee or related persons, which in turn proved to have a role on the sense of job effectiveness and contribution to the firms’ productivity.

Our findings also suggest that workers with higher qualifications (like having a university degree) are also more productive, also bearing in mind that almost 50% of our sample population is college-educated.

## 6. Conclusions

This study provides an integrating model of QWL factors into the trichotomy of (de)motivators of productivity in the workplace. To do so, the original Herzberg’s motivators (workers’ appropriate working time and appropriate salary) and hygiene factors (having a safe work environment and benefiting from occupation healthcare schemes) are integrated as QWL factors, as well as the moderating effects of three burnout factors, i.e., the de-motivators, namely, emotional exhaustion, cynicism and low effectiveness, in order to assess the moderator role of burnout on the relationship between the employee’s QWL and perceptions of their contribution to the organization’s productivity.

Taking as reference the results obtained through estimation of the complete expanded model specification (i.e., model 5), it is worth noting that concerning QWL’s motivators and hygiene factors, such as having an appropriate salary, having a safe work environment and benefiting from occupational healthcare, are confirmed as positively influencing employees’ contribution to productivity. Additionally, new light is shed on the role played by burnout factors, the so-called de-motivators, as a moderator of the relationship between QWL and contribution to productivity. On one hand, the coefficient associated with the interaction term of feeling emotional exhaustion and being appropriately paid denotes a catalyzing effect on the workers’ contribution to productivity. This reveals a positive and significant effect of an appropriate salary on workers feeling burned out, which can act as a compensating catalyst of their contribution to the organization’s productivity. On the other hand, the coefficient noticed on the interaction between emotional exhaustion and having occupational healthcare in the organization shows a negative and significant effect, which can restrict the positive relationship between QWL and contribution to productivity.

The interaction between workers feeling cynics and having a safe work environment also contribute significantly to restricting the relation between QWL and workers’ sense of productivity. Another interesting point is that university education has a significant and positive influence on employees’ contribution to productivity.

The empirical findings now obtained point out that workers are feeling low effectiveness restrict the relationship between the QWL’s motivator factors (such as having an appropriate salary) and their contribution to productivity.

These findings are particularly relevant given the lack of scientific knowledge about the role played by burnout de-motivator factors in absenteeism in the labor market, affecting the relationship between QWL and productivity in organizations. The current findings contribute to advancing knowledge about the interaction between QWL and certain burnout components, which can catalyze or restrict the relationship between QWL and contribution to productivity, guiding the decision-making process and human capital management to foster organizational productivity through QWL, considering burnout levels. This study allows for both theoretical and practical implications in an innovative manner. In the first hand, in terms of theoretical implications, it integrates QWL factors, both motivators and hygiene, into the trichotomy of burnout de-motivators of productivity in the workplace. On the second hand, a set of practical implications can also be derived to leaders and managers that are responsible for designing and implementing innovative occupational health programs in the workplace, for instance: (i) creating the conditions for spurring safe work environments; (ii) designing appropriate occupational healthcare schemes; (iii) creating appropriate salary compensations to balance the stressful working environments; (iv) working on safe job conditions to counterbalance more cynicism attitudes at work; and (v) compensating employees accordingly to promote more job effectiveness, favoring QWL and productivity in the workplace.

The main limitation of this study lies in the impossibility of carrying out a study with a time dimension that could evaluate hypothetical causality relationships between subjective and behavioral components and organizational productivity. Another limitation concerns the response variable representing productivity is based on a subjective measure of the employees’ perception of contribution to productivity.

To expand this topic, deeper research is recommended with a more comprehensive study about the relationship between QWL, burnout, and contribution to productivity, considering different stress environments in the light of turbulent and fast-changing work ecosystems. This would mean working within the organizational context versus distance work (or working from home), contrasting innovative practices of human capital management, including organizational innovation, organizational polyvalence, long-distance project management, creativity labs, organizational hubs, work–life balance and wellness.

## Figures and Tables

**Figure 1 ijerph-18-02425-f001:**
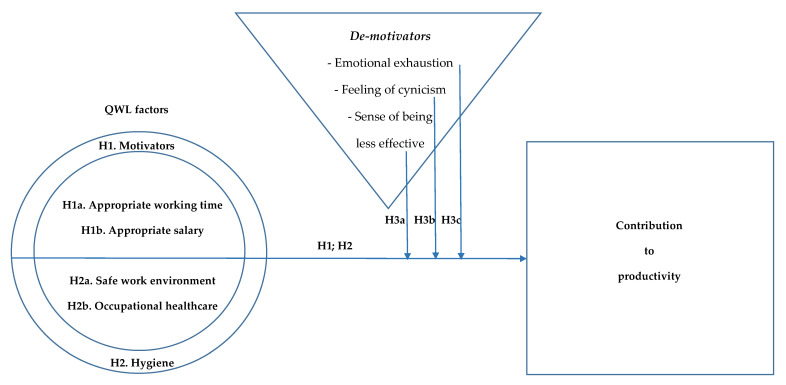
Integrating quality of work life (QWL) factors into the trichotomy of (de)motivators of productivity in the workplace: an operational model of analysis.

**Table 1 ijerph-18-02425-t001:** Sample characterization.

Variables	Type	Weight
Employee gender	Female	47.83
	Male	52.17
Employee age	20–25	8.61
	26–35	34.05
	36–45	36.99
	46–55	13.89
	55+	6.46
Employee marital status	Single	34.83
	Married/Union	58.66
	Other	6.52
Role in organization	Director/Manager	17.77
	Qualified	66.60
	Non-qualified worker	15.63
Employee education	4 years	0.78
	9 years	7.41
	12 years	19.10
	University education	50.49
	Post-graduate	22.22
Organization sector	Primary	1.96
	Secondary	14.29
	Tertiary	77.30
	Public	6.46
Organization size	Micro: 1 to 9	26.37
	Small: 10 to 49	39.06
	Medium: 50 to 249	15.23
	Large: 250 to 1000	13.87
	+Large: 1000+	5.47
Organization age	1 to 6	16.41
	7 to 15	33.98
	16 to 29	25.00
	30 to 49	16.60
	+50	8.01
Employee contract type	Without time limit	67.77
	With time limit	11.13
	Temporary	8.79
	Freelancer	4.49
	Other	7.81
Employees’ position inside an organization	Senior manager	7.25
	Intermediary manager	9.80
	Staff in charge	16.67
	Highly qualified	21.18
	Qualified	24.51
	Semi-qualified	8.24
	Non-qualified	8.63
	Apprentice	2.55
	Do not know	1.18

**Table 2 ijerph-18-02425-t002:** Descriptive statistics and variance inflation factors.

Variable	Obs.	Mean	Std. Dev.	Skewness	Kurtosis	VIF	1/VIF
(1) Contribution to productivity	514	0.8015564	0.3992165	−1.517	0.301	-	-
(2) Appropriate working time	514	0.5019455	0.5004833	−0.008	−2.008	2.08	0.481876
(3) Appropriate salary	514	0.5252918	0.4998464	−0.102	−1.997	1.98	0.506079
(4) Safe work environment	514	0.6536965	0.4762548	−0.648	−1.586	2.20	0.454185
(5) Occupational healthcare	514	0.4922179	0.5004265	0.031	−2.007	2.18	0.458808
(6) Emotional exhaustion	514	0.3715953	0.4837018	0.533	−1.723	5.73	0.174376
(7) Cynicism	514	0.2023346	0.4021317	1.486	0.210	5.34	0.187400
(8) Low effectiveness	514	0.2392996	0.4270716	1.226	−0.500	4.76	0.210019
(9) Female	514	0.4844358	0.5002446	0.062	−2.004	1.06	0.942902
(10) Married	514	0.5603113	0.4968328	−0.244	−1.948	1.11	0.902871
(11) Manager role	514	0.1770428	0.3820768	1.697	0.884	1.11	0.903050
(12) College education	514	0.7256809	0.4466052	−1.015	−0.974	1.08	0.927895
(13) Micro, small and medium-sized	514	0.8035019	0.3977365	−1.532	0.349	1.06	0.946222
(14) Contract without term	514	0.6750973	0.4687947	−0.750	−1.443	1.14	0.880920
					Mean VIF	2.39	

**Table 3 ijerph-18-02425-t003:** Correlation coefficient matrix.

Variables	(1)	(2)	(3)	(4)	(5)	(6)	(7)	(8)	(9)	(10)	(11)	(12)	(13)	(14)
(1) Contribution to productivity	1.0000													
(2) Appropriate working time nt	0.0312	1.0000												
(3) Appropriate salary	0.0447	0.2998 ***	1.0000											
(4) Safe work environment	0.2735 ***	0.0355	−0.0205	1.0000										
(5) Occupational healthcare	0.0801 *	0.2024 ***	0.1722 ***	−0.0277	1.0000									
(6) Emotional exhaustion	0.0091	−0.0151	0.0054	−0.1765 ***	−0.0404	1.0000								
(7) Cynicism	−0.0044	−0.0213	−0.0158	−0.1627 ***	−0.0115	0.3142 ***	1.0000							
(8) Low effectiveness	−0.1097 **	−0.0067	−0.0421	−0.2147 ***	−0.0141	0.1821 ***	0.2283 ***	1.0000						
(9) Female	0.0333	0.0001	0.0873 **	0.0346	−0.0044	0.0360	0.1126 **	0.0585	1.0000					
(10) Married	0.0310	0.0113	−0.0101	0.0143	0.0725	0.0566	0.0071	0.0834 *	−0.0197	1.0000				
(11) Manager role	0.0774 *	−0.0783 *	−0.0694	0.1341 ***	0.0021	0.0336	−0.0433	0.0027	−0.1028 **	0.1131 *	1.0000			
(12) University education	0.2079 ***	0.0416	0.0180	0.1390 ***	−0.0052	−0.0325	−0.0702	−0.0026	−0.0235	0.0527	0.1481 ***	1.0000		
(13) Micro, small and medium sized	−0.0374	−0.0128	0.0201	−0.0718	−0.0714	−0.0149	0.0297	0.0134	0.0091	−0.1421 **	−0.0143	−0.0736 *	1.0000	
(14) Contract without term	0.1235 ***	0.0816 *	0.0143	0.1761 ***	0.0100	0.0005	−0.0849 *	−0.0880 **	−0.0341	0.1889 ***	0.0715	0.1042	−0.1131 *	1.0000

Significance levels: * *p* < 0.10. ** *p* < 0.05. *** *p* < 0.01.

**Table 4 ijerph-18-02425-t004:** Estimation results: basic and expanded models.

	Basic Model Specification	Expanded Model Specifications (with Interaction Terms)			
Dependent Variable: Contribution to Productivity	Model 1	Model 2: Emotional Exhaustion × QWL	Model3: Cynicism × QWL	Model 4: Low Effectiveness × QWL	Model 5: Burnout × QWL
**Independent variables**:					
Appropriate working time	−0.0106251 (0.0356739)	0.002724 (0.0450805)	0.0144145 (0.0402466)	0.0004014 (0.0415648)	0.015954 (0.0486371)
Appropriate salary	0.0248291 (0.0355056)	−0.0531801 (0.0443152)	−0.0046687 (0.0398339)	0.0458615 (0.0409572)	−0.0319938 (0.0475192)
Safe work environment	0.2026167 *** (0.0375959)	0.2264452 *** (0.0478003)	0.2552763 *** (0.0422751)	0.2266892 *** (0.043463)	0.2635709 *** (0.0513709)
Occupational healthcare	0.0689924 ** (0.0345356)	0.1043871 ** (0.0434402)	0.0638563 * (0.0384015)	0.0572153 (0.0397882)	0.0854087 ** (0.0463152)
Emotional exhaustion	0.0465404 (0.037038)	0.0371263 (0.0771303)	0.0538701 (0.0371588)	0.0426644 (0.0372002)	0.0096937 (0.0816304)
Cynicism	0.0495092 (0.0449866)	0.0441092 (0.0449138)	0.1513757* (0.0908087)	0.0562785 (0.0451581)	0.1416273 (0.0947152)
Low effectiveness	−0.0684219 * (0.0413764)	−0.0716486 * (0.0414406)	−0.0695572 * (0.0412559)	0.0237732 (0.0822762)	0.0110992 (0.0842447)
Emot_exhaus × App_work_time		−0.0335582 (0.072789)			−0.0033589 (0.0769073)
Emot_exhaus × Approp_salary		0.2158195 *** (0.0728621)			0.2129359 *** (0.0774649)
Emot_exhaus × Safe_work		−0.0578495 (0.0726563)			−0.0067044 (0.07659)
Emot_exhaus × Occup_health		−0.1050703 (0.0714138)			−0.129529 * (0.0755918)
Cynicism × App_Work_time			−0.1048213 (0.0867841)		−0.0921237 (0.0922984)
Cynicism × Appro_salary			0.1093108 (0.0864103)		0.0712437 (0.0918393)
Cynicism × Safe_work			−0.2130826 ** (0.085264)		−0.1893905 ** (0.0900107)
Cynicism × Occup_health			0.0166369 (0.0860482)		0.037276 (0.0916919)
Low_effectiv × App_Work_time				−0.0493286 (0.0819525)	−0.0254765 (0.0835964)
Low_effectiv × Approp_salary				−0.1035448 (0.0832592)	−0.1828795 ** (0.0852819)
Low_effectiv × Safe_work				−0.0994873 (0.0819374)	−0.0561263 (0.0837742)
Low_effectiv × Occup_health				0.0750905 (0.0824151)	0.1080183 (0.0846726)
Female	0.0218044 (0.0339961)	0.0191741 (0.0338196)	0.0122917 (0.034039)	0.0207438 (0.0340773)	0.0093997 (0.0339436)
Married	0.0024119 (0.0349167)	0.0061544 (0.0348861)	0.0030183 (0.0348193)	−0.0003153 (0.0350419)	0.0025934 (0.0349261)
Manager role	0.0200969 (0.0452851)	0.0284628 (0.0450923)	0.0188389 (0.0451044)	0.0201053 (0.0457597)	0.0223733 (0.0454116)
University education	0.1534784 *** (0.0383223)	0.1526989 *** (0.0382553)	0.1447968 *** (0.0382664)	0.1539299 *** (0.0384025)	0.1469154 *** (0.0383266)
Micro, small and medium sized	0.0056381 (0.0427898)	−0.0016807 (0.042634)	0.0010696 (0.0426245)	0.0038388 (0.0428799)	−0.0062275 (0.0426168)
Permanent contract	0.0513097 (0.0372964)	0.052821 (0.0370996)	0.0537378 (0.0373525)	0.0505762 (0.037627)	0.0542539 (0.0374733)
No. of observations	514	514	514	514	514
R2 (adjusted)	0.1278 ***	0.1461 ***	0.1445 ***	0.1345 ***	0.1676 ***

Standard errors in brackets. Significance levels: * *p* < 0.10. ** *p* < 0.05. *** *p* < 0.01.

## Data Availability

The data presented in this study are available on request from the corresponding author.

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
