# Peer review of "Quality of Work Life and Contribution to Productivity: Assessing the Moderator Effects of Burnout Syndrome"

_ijerph, 2021, doi:10.3390/ijerph18052425_

Round 1
Reviewer 1 Report
Firstly, I would like to thank the authors for the opportunity of reading and reviewing their manuscript. Although the topic presented in the paper is original and fits with the journal scope, it needs some changes to be adapted to the quality of the IJERPH.
Regarding the manuscript, I have some concerns/suggestions:
- Regarding the theoretical framework:
1. Pessimism is not one of the dimensions of Burnout. For example, in line 34, when indicating “emotional exhaustion or energy reduction, the negative emotional state of pessimism and low professional effectiveness [2,4–10]”. I think that by “pessimism” they mean cynicism or depersonalization. As in line 156, they do mention the term correctly. This is a change that should be made as pessimism and cynism are not synonymous. This mistake is made in all the manuscript, even in the abstract.
2. In the first paragraph, when authors say:
“The quality of work-life (QWL) has been associated with higher levels of productivity at the workplace and work-related occupational stress, anxiety, and burnout, which lead to lower job performance and significant costs for organizations [1].” I think this paragraph is misleading since the quality of work-life seems reasonable to be associated with high levels of productivity, but it will be (I understand) the lack of it that is related to stress, anxiety, Burnout, and this lead to increased costs for the organization.
3. From my point of view, the introduction section lacks more fluidity. The authors go from one concept to another without explaining the link between them.
4. When the authors introduce hypothesis 2, I feel lost. There is a prior lack of theoretical development for affirming the hypothesis.
5. In general, I consider that the manuscript lacks a theoretical model or theory that supports the model proposed by the authors. That is, the authors must ask themselves which model can integrate the relationships between the three main concepts they propose, develop them in the theoretical framework, and subsequently explain their results according to it.
- About materials and methods:
6. It is not necessary to include the source after each Table, as the reader understands that the authors made it.
7. The methodology section needs a better structure. The measures and data analysis sections are missing. The procedure is mixed with the description of the sample.
- Regarding, Results and Discussion Section:
8. In my opinion, the manuscript would be better understood if results and discussion were separated.
9. The discussion section is limited to stating whether or not the results are in line with previous studies but does not try to find out why. Again, I defend that if the authors could choose a theoretical model that would help to explain and integrate the main variables, and it would be easier to explain the findings.
- About Conclusions Section:
10. A paragraph regarding theoretical and practical implications is needed
Good luck!
Author Response
Dear Editor-in-Chief of the International Journal of Environmental Research and Public Health,
Prof. Dr. Paul B. Tchounwou
We are very pleased to have the opportunity for resubmitting our co-authored paper titled: Quality of Work Life and Contribution to Productivity: Assessing the moderator effects of Burnout Syndrome.
Firstly, we would like to thank all the reviewers for the constructive feedback and suggestions concerning the previous version of the manuscript. Secondly, we are very pleased to have had the opportunity to revise and resubmit the paper. Considering the responses to the questions raised, we provide a global overview of what was changed according to the review proposals and constructive suggestions made by the reviewers.
Yours faithfully
The Authors
Reviewer 1:
Open Review
(x) I would not like to sign my review report
( ) I would like to sign my review report
English language and style
( ) Extensive editing of English language and style required
( ) Moderate English changes required
( ) English language and style are fine/minor spell check required
(x) I don't feel qualified to judge about the English language and style
|
Yes |
Can be improved |
Must be improved |
Not applicable |
|
|
Does the introduction provide sufficient background and include all relevant references? |
( ) |
( ) |
(x) |
( ) |
|
Is the research design appropriate? |
(x) |
( ) |
( ) |
( ) |
|
Are the methods adequately described? |
( ) |
(x) |
( ) |
( ) |
|
Are the results clearly presented? |
( ) |
( ) |
(x) |
( ) |
|
Are the conclusions supported by the results? |
( ) |
( ) |
(x) |
( ) |
Comments and Suggestions for Authors
Firstly, I would like to thank the authors for the opportunity of reading and reviewing their manuscript. Although the topic presented in the paper is original and fits with the journal scope, it needs some changes to be adapted to the quality of the IJERPH.
Regarding the manuscript, I have some concerns/suggestions:
- Regarding the theoretical framework:
Q1. Pessimism is not one of the dimensions of Burnout. For example, in line 34, when indicating “emotional exhaustion or energy reduction, the negative emotional state of pessimism and low professional effectiveness [2,4–10]”. I think that by “pessimism” they mean cynicism or depersonalization. As in line 156, they do mention the term correctly. This is a change that should be made as pessimism and cynism are not synonymous. This mistake is made in all the manuscript, even in the abstract.
A1: We acknowledge the reviewer’s comment. This was corrected all over the manuscript.
Q2. In the first paragraph, when authors say:
“The quality of work-life (QWL) has been associated with higher levels of productivity at the workplace and work-related occupational stress, anxiety, and burnout, which lead to lower job performance and significant costs for organizations [1].” I think this paragraph is misleading since the quality of work-life seems reasonable to be associated with high levels of productivity, but it will be (I understand) the lack of it that is related to stress, anxiety, Burnout, and this lead to increased costs for the organization.
A2: We acknowledge the reviewer’s comment, which led us to introducing the following revised version of the first paragraph of the Introduction item:
The lack of quality of work life (QWL) is associated with higher levels of work-related occupational stress, anxiety and burnout, which lead to lower job performance and induces to significant costs for organizations [1].
Q3. From my point of view, the introduction section lacks more fluidity. The authors go from one concept to another without explaining the link between them.
A3: We acknowledge the reviewer’s comment, which was taken into account in the revision of the Introduction item, as follows:
The lack of quality of work life (QWL) is associated with higher levels of work-related occupational stress, anxiety and burnout, which lead to lower job performance and induces to significant costs for organizations [1].
Recently, the World Health Organization (WHO) categorized burnout as a job-related phenomenon [2], characterized by chronic stress [3]. Burnout embraces three main dimensions, emotional exhaustion or energy reduction, the negative emotional state or cynicism and low professional effectiveness [2,4–10]. The costs of burnout are growing fast [8,11], affecting currently 13%-25% of the working population [12].
Workers with a high level of emotional intelligence usually have reduced burnout levels [13] as these individuals are more able to deal with stress, which in turn could lead to a higher level of productivity [14]. Emotional regulation techniques can aid individuals to feel accomplished at work [15]. As a sense of low accomplishment is one of the dimensions of burnout, emotional regulation strategies could help to prevent this syndrome.
Such sense of not being able to accomplish duties is associated with the lack of workers’ health and well-being which in turn can be caused by poor working conditions [16]. A related OECD working paper states that health is an important factor in the relationship between work factors and productivity, and strong evidence was found of a negative relationship between job stress and productivity [17].
According to available data, as a whole, from 2018, productivity has been increasing since 1995 in virtually every EU country [18], although its pace growth has been slowing down. Increasing the productivity of individuals and organizations is among the essential objectives of the Europe 2020 Strategy for growth, driven by international competitiveness concerns and the promotion of productivity, growth and sustainability [19], and more recently, with the reinforcement of quality of life through truly green and sustainable growth, as contained in the European Green Deal [20].
Following such need to improve workers’ productivity, the QWL requires further reinforcement in order to spur employees’ motivation, which is increasingly important in the context of digital transition observed in highly skilled, and technologically advanced economies [21]. Adding to the previous statements, the QWL’s improvement is also in line with the worldwide commitment for accomplishing the 17 sustainable development goals (SDGs), as defined by the United Nations.
As previously outlined by [15], besides emotional regulation techniques, there is a need for further research to extend knowledge about burnout components or (de)motivators and their role in accelerating individuals’ personal commitment to organizations’ performance, incorporating the still limited knowledge on the components of QWL as a cornerstone of organizational performance.
In this line of reasoning, the current study provides an innovative assessment of the effects of Burnout as a moderator of the relationship between employees’ QWL and their perceptions of their contribution to the organization’s productivity, highlighting the integration of the QWL Factors into the Trichotomy of (De)Motivators of Productivity in the Work Place. By doing so, it makes a two-fold contribution, namely: (i) testing the moderating effects of burnout on QWL, disaggregating the interaction effects by Motivators and Hygiene per component of QWL; and (ii) revealing the burnout components, as de-motivators, that restrict or catalyze the relationships between distinct components of QWL and employees’ contribution to productivity.
This paper is structured as follows: first, a literature review and the hypotheses are presented, followed by the research methodology. Then the results are discussed and the conclusions drawn, ending with the limitations and implications of the study.
Q4. When the authors introduce hypothesis 2, I feel lost. There is a prior lack of theoretical development for affirming the hypothesis.
A4: We acknowledge the reviewer’s comment. Aiming to conciliate the requests of all reviewers pointing to the need for reinforcing the theoretical basis of the research hypotheses, the following paragraphs were introduced before the set of hypotheses: H1, H1a, H1b; and H2; H2a; H2b.
2.1 Quality of Work Life and Productivity: From the roots into an integrating model
Recovering the roots of the guiding literature on job satisfaction originated in QWL, Herzberg proposed a model where the factors involved in attaining job satisfaction are different from the factors that prompt job dissatisfaction [22][22]. Herzberg asserts that the opposite of job satisfaction is not job dissatisfaction, but instead no job satisfaction [23].
This theory, known as Herzberg’s two factor theory or motivation-hygiene theory proposes that motivation factors are intrinsic to the job while hygiene factors are extrinsic to the job. The motivators, or growth factors are: achievement, recognition, the work itself, responsibility and advancement; while the hygiene or dissatisfaction-avoidance factors are: company policy and administration, supervision, interpersonal relationships, working conditions, salary, status and security [22,23]. The results of Herzberg’s work indicate that motivators are the cause of job satisfaction and hygiene factors the cause of unhappiness on the job [23]. This theory was the subject of many scientific studies, some supporting this theory[24–28], while others counter it [29–33].
Moreover, more recently another theory arose, the trichotomy of motivator factors in the workplace [34], based upon Herzberg’s theory [22] and also the theory of tourist motivation factors [35]. The trichotomy of motivator factors adds another factor to Herzberg’s two-factor theory and identifies 3 factors involved in job satisfaction: motivators, hygiene factors and de-motivators. This theory identifies as motivators for job satisfaction the following factors: bonuses, promotion opportunities, personal development opportunities, flextime, cafeteria benefits, recognizing merit and training paid by the employer [34]. As hygiene factors, Koziol et al. identify: compensation, working hours, workload, interpersonal relations, friendly atmosphere at work, industrial safety, work content, company policy, responsibility and social scheme activities. For the final and added de-motivator factors, the following are described: mobbing by superiors/co-workers, stress at work, work exceeding employee’s psychophysical potential and qualifications, short-term contracts, employer’s continuous and close supervision and lack of possibility of changing status quo/making improvements [34]. The author assumes the motivation factors represent the stimulants, the hygiene factors represent the nominants while the de-motivator factors represent the denominants [34]. In addition, the author considers that in order to make improvements to the motivation system, the de-motivators (denominants) should be eliminated, the hygiene factors (nominants) should be optimized and lastly the motivators (stimulants) should be maximized [34].
Whit the motivation of designing an integrating model of the QWL Factors into the Trichotomy of (De)Motivators of Productivity in the Work Place, it should be stressed that albeit there is a vast amount of literature on the subject of QWL, many researchers agree that QWL is different from job satisfaction and that it deals with employees’ well-being [36–40].
Q5. In general, I consider that the manuscript lacks a theoretical model or theory that supports the model proposed by the authors. That is, the authors must ask themselves which model can integrate the relationships between the three main concepts they propose, develop them in the theoretical framework, and subsequently explain their results according to it.
A5: We acknowledge the reviewer’s comment. For addressing it, a substantial revision effort was put into action by integrating the original Herzberg’s approach plus the trichotomy of motivator factors, as presented before, in the following paragraphs:
Recovering the roots of the guiding literature on job satisfaction originated in QWL, Herzberg proposed a model where the factors involved in attaining job satisfaction are different from the factors that prompt job dissatisfaction [22][22]. Herzberg asserts that the opposite of job satisfaction is not job dissatisfaction, but instead no job satisfaction [23].
This theory, known as Herzberg’s two factor theory or motivation-hygiene theory proposes that motivation factors are intrinsic to the job while hygiene factors are extrinsic to the job. The motivators, or growth factors are: achievement, recognition, the work itself, responsibility and advancement; while the hygiene or dissatisfaction-avoidance factors are: company policy and administration, supervision, interpersonal relationships, working conditions, salary, status and security [22,23]. The results of Herzberg’s work indicate that motivators are the cause of job satisfaction and hygiene factors the cause of unhappiness on the job [23]. This theory was the subject of many scientific studies, some supporting this theory[24–28], while others counter it [29–33].
Moreover, more recently another theory arose, the trichotomy of motivator factors in the workplace [34], based upon Herzberg’s theory [22] and also the theory of tourist motivation factors [35]. The trichotomy of motivator factors adds another factor to Herzberg’s two-factor theory and identifies 3 factors involved in job satisfaction: motivators, hygiene factors and de-motivators. This theory identifies as motivators for job satisfaction the following factors: bonuses, promotion opportunities, personal development opportunities, flextime, cafeteria benefits, recognizing merit and training paid by the employer [34]. As hygiene factors, Koziol et al. identify: compensation, working hours, workload, interpersonal relations, friendly atmosphere at work, industrial safety, work content, company policy, responsibility and social scheme activities. For the final and added de-motivator factors, the following are described: mobbing by superiors/co-workers, stress at work, work exceeding employee’s psychophysical potential and qualifications, short-term contracts, employer’s continuous and close supervision and lack of possibility of changing status quo/making improvements [34]. The author assumes the motivation factors represent the stimulants, the hygiene factors represent the nominants while the de-motivator factors represent the denominants [34]. In addition, the author considers that in order to make improvements to the motivation system, the de-motivators (denominants) should be eliminated, the hygiene factors (nominants) should be optimized and lastly the motivators (stimulants) should be maximized [34].
In addition, following the highly valuable set of suggestions of the reviewers, a new integrative and operational model of analysis was (re)designed, as well as the research hypotheses, which are presented in the new Figure 1:
Figure 1. Integrating QWL Factors into the Trichotomy of (De)Motivators of Productivity in the Work Place: An Operational Model of Analysis
- About materials and methods:
Q6. It is not necessary to include the source after each Table, as the reader understands that the authors made it.
A6: We acknowledge the reviewer’s comment. All the sources mentioning: Own elaboration; were removed.
Q7. The methodology section needs a better structure. The measures and data analysis sections are missing. The procedure is mixed with the description of the sample.
A7: We acknowledge the reviewer’s comment. For addressing this comment, the structure of the methodology section was revised, in the following terms:
- Methodology
3.1. Sample
3.2. Measures and Preliminary Data Analysis
3.3. Model specification
- Regarding, Results and Discussion Section:
Q8. In my opinion, the manuscript would be better understood if results and discussion were separated.
A8: We acknowledge the reviewer’s comment. For addressing this comment, the structure of the manuscript was revised, in the following terms:
- Results
- Discussion
- Conclusions
Q9. The discussion section is limited to stating whether or not the results are in line with previous studies but does not try to find out why. Again, I defend that if the authors could choose a theoretical model that would help to explain and integrate the main variables, and it would be easier to explain the findings.
A9: We acknowledge the reviewer’s comment. For addressing this comment, the item 5. Discussion was rewritten, discussing the findings and integrating the original Herzberg’s approach plus the trichotomy of motivator factors, in the following terms:
- Discussion
According to the results obtained, all OLS models (1-5) give support to H2, stating that the QWL’s hygiene factors have a positive and significant effect on the contribution to productivity, and this is in line with previous literature [53], which states that a safer work environment will make employees perform at their best level, also in line with the previous findings of Kiriago and Bwisa (2013) and Leitão et al. (2019) [50,52], who found a negative correlation between a poor work environment and QWL. This agrees with previous work that considers health an important component of QWL [42,46,48]. Furthermore, as Herzberg’s work shows not having balanced hygiene factors causes no job satisfaction [50]. In addition, the results obtained are aligned with prior theories suggesting that the hygiene factors (i.e. nominants) should be optimized in order to achieve satisfaction on the job [61], being the later associated with employees’ sense of contribution to productivity. This is also aligned with previous works defending that QWL can be expressed through the satisfaction of a set of employee needs, which is associated with their participation, contribution, at the workplace [29]. No significant direct effects related to the hypothesis 1, were found.
According to the results obtained with model 1, the H2a and H2b cannot be rejected. Notably, regarding the Herzberg’s hygiene factors, having a safe work environment and benefiting of occupational healthcare schemes is able to catalyze the contribution to productivity. The same support is found in models 2, 3 and 5. In model 4, it is found support only for H2a.
The current study raised and tested a set of hypothesis around the Burnout’s de-motivator factors (that is, emotional exhaustion, cynicism, and sense of low effectiveness) and their moderator role on the relationship between the QWL’s motivators and hygiene factors, and the contribution to productivity. In fact, it is found support for H3a (in models 2 and 5), H3b and H3c. (in models 3 and 5). Indeed, benefiting from an appropriate salary, schemes of occupational health (QWL’s motivators) at work coupled with emotional exhaustion restricts the relationship between QWL and contribution to productivity (H3a), which is aligned with previous work [11,68,71,72,82,91]. Highly stressful working environments is associated with a higher burnout, being staff’ stress a positive predictor of burnout, as prior studies already conveyed [2,3,66,85]. Both emotional exhaustion and burnout impact negatively on the worker’s productivity [86], being extreme forms of stress, causing energy depletion [87], and thus slowing productivity [88,89,90]. The results now obtained ratify previous findings, connecting QWL’s motivators and hygiene factors with emotional exhaustion (the primary dimension of burnout), pointing out a negative impact of burnout on productivity.
Moreover, the empirical evidence now obtained, signals that cynicism, a form of pessimism, restricts the relationship between the QWL’s motivators and hygiene factors, and the contribution to productivity, thus supporting H3b. Such results are in line with prior literature, which addressed the relationship between optimism/cynicism at work and productivity, concluding that optimist workers are more productive at work than pessimists [91]. In addition, cynic and pessimist workers are more absent than optimists are. The current study underpins a negative and significant effect of the Burnout’s de-motivator factor, cynicism, as well as its moderator role on the relationship between the QWL’s hygiene factors (such as, having a safe workplace), and the contribution to productivity (as found in models 3 and 5). Optimistic workers affect positively the effect of hygiene factors, such as safe work environments, on the sense of job productivity.
Taking as reference the results obtained in model 5, there is support for H3c, as there is evidence of a negative and significant effect of the Burnout’s de-motivator factor, low effectiveness of the workers, and its moderator role on the relationship between the QWL’s motivators (such as, having an appropriate salary), and the contribution to productivity. This is aligned with previous literature [91,92] that advocated a positive influence of personal accomplishment on productivity and suggesting that workers that feel low personal effectiveness, this associated with burnout, are less productive on the job. The empirical findings now obtained also unveils that low effectiveness when combined with appropriate salary restricts the contribution to productivity. These results highlight the importance of implementing a set of initiatives for fighting job related burnout, as in terms of costs it is estimated that it affects between 13 and 25% of working population [8,11], influencing employees’ performance and thus firms’ productivity. From the current findings, it is also possible to provide lines of action, namely designing appropriate payment schemes, including benefits programmes targeted to personal expenses cut for the employee or related persons, which in turn proved to have a role on the sense of job effectiveness and contribution to the firms’ productivity. Our findings also suggest that workers with higher qualifications (like having a university degree) are also more productive, bearing also in mind that almost 50% of our sample population is college educated.
- About Conclusions Section:
Q10. A paragraph regarding theoretical and practical implications is needed.
A10: We acknowledge the reviewer’s comment. For addressing this comment, in the item 6. Conclusions, the following sentences were added:
This study provides an integrating model of QWL Factors into the Trichotomy of (De)Motivators of Productivity in the workplace. To do so, the original Herzberg’s motivators (workers’ appropriate working time and appropriate salary) and hygiene factors (having a safe work environment and benefiting from occupation healthcare schemes) are integrated as QWL factors, as well as the moderating effects of three burnout factors, i.e., the de-motivators, namely, emotional exhaustion, cynicism and low effectiveness, in order to assess the moderator role of burnout on the relationship between the employee’s QWL and perceptions of their contribution to the organization’s productivity.
Taking as reference the results obtained through estimation of the complete expanded model specification (i.e. model 5), it is worth noting that concerning QWL’s motivators and hygiene factors, such as having an appropriate salary, having a safe work environment and benefiting from occupational healthcare, are confirmed as positively influencing employees’ contribution to productivity. Additionally, new light is shed on the role played by burnout factors, the so-called de-motivators, as a moderator of the relationship between QWL and contribution to productivity. On the one hand, the coefficient associated with the interaction term of feeling emotional exhaustion and being appropriately payed, denotes a catalyzing effect on the workers’ contribution to productivity. This reveals a positive and significant effect of an appropriate salary on workers feeling burned out, which can act as a compensating catalyst of their contribution to the organization’s productivity. On the other hand, the coefficient noticed on the interaction between emotional exhaustion and having occupational healthcare in the organization shows a negative and significant effect, which can restrict the positive relationship between QWL and contribution to productivity.
The interaction between workers feeling cynics and having a safe work environment also contribute significantly to restricting the relation between QWL and workers’ sense of productivity. Another interesting point is that a university education has a significant and positive influence on employees’ contribution to productivity.
The empirical findings now obtained point out that workers feeling low effective restrict the relationship between the QWL’s motivator factors (such as having an appropriate salary) and their contribution to productivity.
These findings are particularly relevant given the lack of scientific knowledge about the role played by burnout de-motivator factors in absenteeism in the labor market affecting the relationship between QWL and productivity in organizations. The current findings contribute to advancing knowledge about the interaction between QWL and certain burnout components, which can catalyze or restrict the relationship between QWL and contribution to productivity, guiding the decision-making process and human capital management to foster organizational productivity through QWL, considering burnout levels. This study allows for both theoretical and practical implications in an innovative manner. In the first hand, in terms of theoretical implications, it integrates QWL factors, both motivators and hygiene, into the trichotomy of burnout de-motivators of productivity in the work place. In the second hand, a set of practical implications can also be derived to leaders and managers that are responsible for designing and implementing innovative occupational health programs in the workplace, for instance: (i) creating the conditions for spurring safe work environments; (ii) designing appropriate occupational healthcare schemes; (iii) creating appropriate salary compensations to balance the stressful working environments; (iv) working on safe job conditions to counterbalance more cynicism attitudes at work; and (v) compensating employees accordingly to promote more job effectiveness, favoring QWL and productivity in the workplace.

Reviewer 2 Report
The article presents the relationship between Quality of Work Life QWL and productivity in the workplace. The authors correctly characterized QWL elements. It was assumed that some of them have a positive and some negative impact on labor productivity. This is a simplification. The mechanisms of the impact of such factors on motivation and individual work productivity have been described on the basis of the theory of motivation. I propose to use the classical achievements of Hertzberg (two-factor theory) and its extension towards trichotomy. I think that such a systematization of QWL elements will be beneficial for the first part of the article.
Author Response
Dear Editor-in-Chief of the International Journal of Environmental Research and Public Health,
Prof. Dr. Paul B. Tchounwou
We are very pleased to have the opportunity for resubmitting our co-authored paper titled: Quality of Work Life and Contribution to Productivity: Assessing the moderator effects of Burnout Syndrome.
Firstly, we would like to thank all the reviewers for the constructive feedback and suggestions concerning the previous version of the manuscript. Secondly, we are very pleased to have had the opportunity to revise and resubmit the paper. Considering the responses to the questions raised, we provide a global overview of what was changed according to the review proposals and constructive suggestions made by the reviewers.
Yours faithfully
The Authors
Reviewer 2:
Open Review
(x) I would not like to sign my review report
( ) I would like to sign my review report
English language and style
( ) Extensive editing of English language and style required
( ) Moderate English changes required
( ) English language and style are fine/minor spell check required
(x) I don't feel qualified to judge about the English language and style
|
Yes |
Can be improved |
Must be improved |
Not applicable |
|
|
Does the introduction provide sufficient background and include all relevant references? |
( ) |
(x) |
( ) |
( ) |
|
Is the research design appropriate? |
(x) |
( ) |
( ) |
( ) |
|
Are the methods adequately described? |
(x) |
( ) |
( ) |
( ) |
|
Are the results clearly presented? |
(x) |
( ) |
( ) |
( ) |
|
Are the conclusions supported by the results? |
(x) |
( ) |
( ) |
( ) |
Comments and Suggestions for Authors
Q1: The article presents the relationship between Quality of Work Life QWL and productivity in the workplace. The authors correctly characterized QWL elements. It was assumed that some of them have a positive and some negative impact on labor productivity. This is a simplification. The mechanisms of the impact of such factors on motivation and individual work productivity have been described on the basis of the theory of motivation. I propose to use the classical achievements of Hertzberg (two-factor theory) and its extension towards trichotomy. I think that such a systematization of QWL elements will be beneficial for the first part of the article.
A1: We acknowledge the reviewer’s comment. For addressing it, a substantial revision effort was put into action by integrating the original Herzberg’s approach plus the trichotomy of motivator factors, as presented before, in the following paragraphs:
Recovering the roots of the guiding literature on job satisfaction originated in QWL, Herzberg proposed a model where the factors involved in attaining job satisfaction are different from the factors that prompt job dissatisfaction [22][22]. Herzberg asserts that the opposite of job satisfaction is not job dissatisfaction, but instead no job satisfaction [23].
This theory, known as Herzberg’s two factor theory or motivation-hygiene theory proposes that motivation factors are intrinsic to the job while hygiene factors are extrinsic to the job. The motivators, or growth factors are: achievement, recognition, the work itself, responsibility and advancement; while the hygiene or dissatisfaction-avoidance factors are: company policy and administration, supervision, interpersonal relationships, working conditions, salary, status and security [22,23]. The results of Herzberg’s work indicate that motivators are the cause of job satisfaction and hygiene factors the cause of unhappiness on the job [23]. This theory was the subject of many scientific studies, some supporting this theory[24–28], while others counter it [29–33].
Moreover, more recently another theory arose, the trichotomy of motivator factors in the workplace [34], based upon Herzberg’s theory [22] and also the theory of tourist motivation factors [35]. The trichotomy of motivator factors adds another factor to Herzberg’s two-factor theory and identifies 3 factors involved in job satisfaction: motivators, hygiene factors and de-motivators. This theory identifies as motivators for job satisfaction the following factors: bonuses, promotion opportunities, personal development opportunities, flextime, cafeteria benefits, recognizing merit and training paid by the employer [34]. As hygiene factors, Koziol et al. identify: compensation, working hours, workload, interpersonal relations, friendly atmosphere at work, industrial safety, work content, company policy, responsibility and social scheme activities. For the final and added de-motivator factors, the following are described: mobbing by superiors/co-workers, stress at work, work exceeding employee’s psychophysical potential and qualifications, short-term contracts, employer’s continuous and close supervision and lack of possibility of changing status quo/making improvements [34]. The author assumes the motivation factors represent the stimulants, the hygiene factors represent the nominants while the de-motivator factors represent the denominants [34]. In addition, the author considers that in order to make improvements to the motivation system, the de-motivators (denominants) should be eliminated, the hygiene factors (nominants) should be optimized and lastly the motivators (stimulants) should be maximized [34].
The research hypotheses were revised accordingly with the following:
Hypothesis 1 (H1): The QWL’s motivators factors have a positive relationship with the contribution to productivity.
Hypothesis 1a (H1a): The appropriate working time has a positive and significant effect on the contribution to productivity.
Hypothesis 1b (H1b): The appropriate salary has a positive and significant effect on the contribution to productivity.
Hypothesis 2 (H2): The QWL’s hygiene factors have a positive relationship with the contribution to productivity.
Hypothesis 2a (H2a): The safe work environment has a positive and significant effect on the contribution to productivity.
Hypothesis 2b (H2b): The occupational healthcare has a positive and significant effect on the contribution to productivity.
Hypothesis 3 (H3): The Burnout’s de-motivators factors moderate the relationship between the QWL’s motivators and hygiene factors, and the contribution to productivity.
Hypothesis 3a (H3a): Emotional exhaustion restricts the relationship between the QWL’s motivators and hygiene factors, and the contribution to productivity.
Hypothesis 3b (H3b): The feeling of cynicism restricts the relationship between the QWL’s motivators and hygiene factors, and the contribution to productivity.
Hypothesis 3c (H3c): The sense of being less effective restricts the relationship between the QWL’s motivators and hygiene factors, and the contribution to productivity.
In addition, following the highly valuable set of suggestions of the reviewers, a new integrative and operational model of analysis was (re)designed, as well as the research hypotheses, which are presented in the new Figure 1:
Figure 1. Integrating QWL Factors into the Trichotomy of (De)Motivators of Productivity in the Work Place: An Operational Model of Analysis

Reviewer 3 Report
I comend the authors for taking time out to look out the subject. To help improve the work quality and increase potential readers interest the following suggestions have been advanced:
- The abstract has not provide insight to method adopted in the study but only inform of the outcome.
- Line 295-298, there was reference to interview of employees however moving forward in line 302-304 questionnaire was mentioned as tool used to sample the participants that informed the study outcome. The method used need clarity
- Part of Section 3.1 (line 308-330) and Table 1 are presentation of findings. It is not clear why this was included in the methodology section. This should be moved to the result section of the paper.
- Tables 2 and 3 and their corresponding text should likewise be moved to the result section
- Table 4 should be moved to the result section as well.
- Result and discussion heading in my opinion should read discussion as the results are presented in sections mentioned above already.
- Overall, three sections: methodology, result and discussion of the paper need restructuring to ensure that each achieved its intended objectives.
Author Response
Dear Editor-in-Chief of the International Journal of Environmental Research and Public Health,
Prof. Dr. Paul B. Tchounwou
We are very pleased to have the opportunity for resubmitting our co-authored paper titled: Quality of Work Life and Contribution to Productivity: Assessing the moderator effects of Burnout Syndrome.
Firstly, we would like to thank all the reviewers for the constructive feedback and suggestions concerning the previous version of the manuscript. Secondly, we are very pleased to have had the opportunity to revise and resubmit the paper. Considering the responses to the questions raised, we provide a global overview of what was changed according to the review proposals and constructive suggestions made by the reviewers.
Yours faithfully
The Authors
Reviewer 3:
Open Review
(x) I would not like to sign my review report
( ) I would like to sign my review report
English language and style
( ) Extensive editing of English language and style required
( ) Moderate English changes required
(x) English language and style are fine/minor spell check required
( ) I don't feel qualified to judge about the English language and style
|
Yes |
Can be improved |
Must be improved |
Not applicable |
|
|
Does the introduction provide sufficient background and include all relevant references? |
(x) |
( ) |
( ) |
( ) |
|
Is the research design appropriate? |
( ) |
(x) |
( ) |
( ) |
|
Are the methods adequately described? |
( ) |
( ) |
(x) |
( ) |
|
Are the results clearly presented? |
( ) |
( ) |
(x) |
( ) |
|
Are the conclusions supported by the results? |
( ) |
(x) |
( ) |
( ) |
Comments and Suggestions for Authors
I comend the authors for taking time out to look out the subject. To help improve the work quality and increase potential readers interest the following suggestions have been advanced:
Q1: The abstract has not provided insight to method adopted in the study but only inform of the outcome.
A1: We acknowledge the reviewer’s comment, which was addressed by revising the abstract, mentioning the empirical method in use, that is, an OLS multiple regression, with interaction terms, accordingly to the following:
Abstract: This study is focused on assessing the effects of Burnout as a moderator of the relationship between employees’ quality of work life (QWL) and their perceptions of their contribution to the organization’s productivity, by integrating the QWL Factors into the Trichotomy of (De)Motivators of Productivity in the Work Place. The empirical findings resulting from an OLS multiple regression, with interaction terms, applied to a survey administered at 514 employees in 6 European countries, point out two important insights: (i) QWL hygiene factors (e.g. safe work environment, and occupational healthcare) positively and significantly influence the contribution to productivity; and (ii) burnout de-motivator factors (that is, low effectiveness, cynicism, and emotional exhaustion) significantly moderate the relationship between QWL and the contribution to productivity. Combining burnout with other QWL components, such as occupational health, safe work, and appropriate salary, new insights are provided concerning the restricting (i.e., low effectiveness, and cynicism) and catalyzing (emotional exhaustion) burnout components of contribution to productivity. These findings are particularly relevant given the increased weight of burnout, mental disorders and absenteeism in the labor market, affecting individuals’ quality of life and organizations’ performance and costs.
Q2: Line 295-298, there was reference to interview of employees however moving forward in line 302-304 questionnaire was mentioned as tool used to sample the participants that informed the study outcome. The method used need clarity
A2: We acknowledge the reviewer’s comment, which was addressed by presenting in item 3.1. Sample, the summing of the desk research method in use, in the following sentences:
3.1. Sample
The study comprehends the analysis of the responses to a survey funded on different questionnaires previously used to carry out related surveys on health and wellbeing in the workplace, including the pioneering measure on quality of work life developed by Sirgy et al. (2001), and the set of analytical tools surveyed and empirically operationalized by Leitão et al. (2019).
The survey was conducted from April to July 2018. A total of twelve project partners originating from Italy, Bulgaria, Cyprus, Portugal, Greece and Spain participated in data collection, by interviewing employees. The intention was not to interview company owners or general managers to avoid bias in the responses. A convenience sample based on a random selection procedure was used. In each organization, a contact person was identified to ensure completion of the questionnaire, which was afterwards validated by the research team.
The questionnaires were applied through personal interviews to ensure a maximum response rate. The partners followed a set of instructions for selecting interviewees: 15 companies among micro, small and medium-sized firms (10% of interviewees for each category—EU definition of SME), plus 5 among large firms and public entities, involving two employees per organization and totaling 514 questionnaires.
Q3: Part of Section 3.1 (line 308-330) and Table 1 are presentation of findings. It is not clear why this was included in the methodology section. This should be moved to the result section of the paper.
A3: We acknowledge the reviewer’s comment. For conciliating the reviewers’ requests, this is presented as part of the sub-section: 3.2. Measures and Preliminary Data Analysis.
Q4: Tables 2 and 3 and their corresponding text should likewise be moved to the result section
A4: We acknowledge the reviewer’s comment. Following the same rationale, in order to conciliate the reviewers’ requests, this is also presented as part of the sub-section: 3.2. Measures and Preliminary Data Analysis.
Q5: Table 4 should be moved to the result section as well.
A5: We acknowledge the reviewer’s comment. Following this recommendation, Table 4 is included in the item 4. Results.
Q6: Result and discussion heading in my opinion should read discussion as the results are presented in sections mentioned above already.
A6: We acknowledge the reviewer’s comment. For conciliating the reviewers’ requests and addressing this comment, the structure of the manuscript was revised, in the following terms:
- Results
- Discussion
- Conclusions
Q7: Overall, three sections: methodology, result and discussion of the paper need restructuring to ensure that each achieved its intended objectives.
A7: We acknowledge the reviewer’s comment. For addressing this highly valuable comment, all the sections identified were revised according to the recommendations received, integrating the original Herzberg’s approach plus the trichotomy of motivator factors, as previously presented.

Round 2
Reviewer 1 Report
I would like to thanks the authors for properly addressing my suggestions. Congrats!